# Feasibility of Real-Time Central Surgical Review for Patients with Advanced-Stage Hepatoblastoma in the JPLT3 Trial

**DOI:** 10.3390/children9020234

**Published:** 2022-02-10

**Authors:** Tomoro Hishiki, Shohei Honda, Yuichi Takama, Yukihiro Inomata, Hideaki Okajima, Ken Hoshino, Tatsuya Suzuki, Ryota Souzaki, Motoshi Wada, Mureo Kasahara, Koichi Mizuta, Takaharu Oue, Akiko Yokoi, Takuro Kazama, Shugo Komatsu, Isamu Saeki, Osamu Miyazaki, Tetsuya Takimoto, Kohmei Ida, Kenichiro Watanabe, Eiso Hiyama

**Affiliations:** 1Department of Pediatric Surgery, Chiba University Graduate School of Medicine, 1-8-1 Inohana, Chuo-ku, Chiba 260-8670, Chiba, Japan; skomatsu@chiba-u.jp; 2Department of Gastroenterological Surgery I, Hokkaido University Hospital, Sapporo 060-8648, Hokkaido, Japan; s-honda@med.hokudai.ac.jp; 3Department of Pediatric Surgery, Osaka City General Hospital, Osaka 534-0021, Osaka, Japan; takama@pedsurg.med.osaka-u.ac.jp; 4Kumamoto Rosai Hospital, Yatsushiro 866-8533, Kumamoto, Japan; yino@kuh.kumamoto-u.ac.jp; 5Department of Pediatric Surgery, Kanazawa Medical University, Kanazawa 920-0293, Ishikawa, Japan; hokajima@kanazawa-med.ac.jp; 6Department of Pediatric Surgery, Keio School of Medicine, Keio University, Tokyo 108-8345, Tokyo, Japan; hoshino@z7.keio.jp; 7Department of Pediatric Surgery, Fujita Health University Hospital, Toyoake 470-1192, Aichi, Japan; tsuzuki@fujita-hu.ac.jp; 8Department of Pediatric Surgery, Graduate School of Medical Sciences, Kyushu University, Fukuoka 812-8582, Fukuoka, Japan; ryotas@med.kyushu-u.ac.jp; 9Department of Pediatric Surgery, Tohoku University School of Medicine, Sendai 980-8574, Miyagi, Japan; wada@ped-surg.med.tohoku.ac.jp (M.W.); kazama@ped-surg.med.tohoku.ac.jp (T.K.); 10Organ Transplantation Center, National Center for Child Health and Development, Tokyo 157-8535, Tokyo, Japan; kasahara-m@ncchd.go.jp; 11Transplant Center, Saitama Children’s Medical Center, Saitama 330-8777, Saitama, Japan; koimizu@jichi.ac.jp; 12Department of Pediatric Surgery, Hyogo College of Medicine, Nishinomiya 663-8501, Hyogo, Japan; ta-oue@hyo-med.ac.jp; 13Department of Pediatric Surgery, Kobe Children’s Hospital, Kobe 650-0047, Hyogo, Japan; yokoi_kch@hp.pref.hyogo.jp; 14Department of Pediatric Surgery, Hiroshima University Hospital, Hiroshima 734-8551, Hiroshima, Japan; isaeki@hiroshima-u.ac.jp (I.S.); eiso@hiroshima-u.ac.jp (E.H.); 15Department of Diagnostic Radiology, National Center for Child Health and Development, Tokyo 157-8535, Tokyo, Japan; miyazaki-o@ncchd.go.jp; 16Department of Childhood Cancer Data Management, National Center for Child Health and Development, Tokyo 157-8535, Tokyo, Japan; takimoto-t@ncchd.go.jp; 17Department of Pediatrics, Teikyo University Mizonokuchi Hospital, Kawasaki 213-8507, Kanagawa, Japan; komei0313@med.teikyo-u.ac.jp; 18Department of Hematology and Oncology, Shizuoka Children’s Hospital, Shizuoka 420-8660, Shizuoka, Japan; k-watanabe@i.shizuoka-pho.jp; 19Natural Science Center for Basic Research and Development, Hiroshima University, Hiroshima 734-8551, Hiroshima, Japan

**Keywords:** hepatoblastoma, clinical trial, surgery, central review, cloud-based remote image viewing system, liver transplantation

## Abstract

In the JPLT3 study, a real-time central surgical reviewing (CSR) system was employed aimed at facilitating early referral of candidates for liver transplantation (LTx) to centers with pediatric LTx services. The expected consequence was surgery, including LTx, conducted at the appropriate time in all cases. This study aimed to review the effect of CSR on institutional surgical decisions in cases enrolled in the JPLT3 study. Real-time CSR was performed in cases in which complex surgeries were expected, using images obtained after two courses of preoperative chemotherapy. Using the cloud-based remote image viewing system, an expert panel consisting of pediatric and transplant surgeons reviewed the images and commented on the expected surgical strategy or the necessity of transferring the patient to a transplant unit. The results were summarized and reported to the treating institutions. A total of 41 reviews were conducted for 35 patients, and 16 cases were evaluated as possible candidates for LTx, with the treating institutions being advised to consult a transplant center. Most of the reviewed cases promptly underwent definitive liver surgeries, including LTx per protocol.

## 1. Introduction

Hepatoblastoma (HB) is the most common malignant liver tumor in children, and accounts for approximately 1% of all pediatric malignancies [1,2]. The prognosis of patients with HB has dramatically increased in the last three decades, mainly owing to the application of cisplatin-based neoadjuvant and adjuvant chemotherapies [3,4,5,6]. However, since complete resection of the tumor is critical for the patient’s long-term survival, surgery remains the mainstay of treatment for HBs. Recent developments in surgical treatment strategies, including extreme hepatic resection techniques and liver transplantation (LTx) with safety and preserved liver function, have also contributed to the improved survival of patients [7,8,9,10]. Currently, tumors that remain unresectable with hepatectomy after neoadjuvant chemotherapy are generally candidates for LTx.

The Japanese Study Group for Pediatric Liver Tumors (JPLT; currently the Liver Tumor Committee of the Japan Children’s Cancer Group) conducted a multicenter prospective study from 1999 to 2012 (JPLT-2) [3,11]. Unresectable tumors at diagnosis were treated with neoadjuvant chemotherapy, which was intensified according to the response to first-line treatment. Chemotherapy was repeated for up to six courses if surgical treatment was unfeasible. Of the 361 patients, 5 died without receiving any definitive surgery, and 12 remained unresectable after the protocol treatment chemotherapy. It was not until 2010 that LTx was covered by insurance in our country. Thus, most patients enrolled in this study, especially in the first 10 years of the study, did not have access to LTx [11]. Patients remaining unresectable after chemotherapy typically receive excessive courses of chemotherapy seeking maximum shrinkage of the tumor, which did not effectively improve the resectability of the tumor, but resulted in increased chemoresistance and chemotoxicity [11].

The JPLT3 study consisted of three risk-adapted prospective trials for HB, subclassifying patients into standard-, intermediate-, and high-risk (named JPLT3-S, JPLT3-I, and JPLT3-H, respectively). In response to the significant delay of surgery in cases enrolled in the JPLT-2, a real-time central surgical review platform using a cloud-based remote image viewing system was developed to facilitate the appropriate surgical procedure to be performed at proper timing. A panel consisting of experts in liver surgery or LTx voluntarily participated in the review.

In this study, the surgical review process conducted in cases enrolled in the JPLT3 trial was retrospectively reviewed. We focused on the timeliness of surgical reviews to assist with treating institutions in deciding whether the patient should be referred to a facility capable of LTx and extreme liver surgeries.

## 2. Materials and Methods

The detailed treatment protocol and outcome of the high-risk feasibility study (JPLT3-H) are published elsewhere [12]. Details of treatment and outcome in standard-risk and intermediate-risk patients (JPLT3-S, JPLT3-I) are to be published after the designated follow-up is completed. Risk group classification was defined according to the tentative results obtained from the Children’s Hepatic tumor International Collaboration (CHIC) database analysis during the protocol design [13,14]. Briefly, the high-risk group consisted of cases with M1 or N2 PRETEXT annotations [15], or cases with serum alpha-fetoprotein (AFP) level of <100 ng/dL at diagnosis. Intermediate-risk included non-high-risk patients with either of the following: PRETEXT IV tumor; annotation factors E1, E1s, E2, E2a, H1, N1, P2, P2a, V3, or V3a; multifocal; or age > 3 at diagnosis. Standard-risk patients included all other HBs. The JPLT3-S protocol treatment consisted of six courses of cisplatin monotherapy and surgery, in which 80 mg/m^2^ cisplatin was administered over 24 h every 14 days, as in the SIOPEL-3 standard-risk study [16]. The JPLT3-I protocol treatment consisted of six courses of the combination of 80 mg/m^2^ cisplatin administered over 24 h on day one and 60 mg/m^2^ doxorubicin administered over 48 h on days 2 and 3, following the PLADO regimen published previously [17]. Surgical resection of the primary tumor could be performed at any time after four courses of chemotherapy in the JPLT3-I and -S. The JPLT3-H assessed the feasibility and safety of the SIOPEL-4 regimen in a Japanese patient cohort; thus, the protocol treatment regimen was nearly identical to that of SIOPEL-4 [4]. The surgical guidelines indicated the following cases as possible candidates for LTx and recommended early consultation with transplant centers: (1) multifocal PRETEXT IV tumor; (2) unifocal PRETEXT IV tumor without downstaging; and (3) centrally located tumors with massive invasion to the portal bifurcation or the hepatic veins that are unlikely to be resected safely with conventional hepatectomies.

In all studies, radiological evaluation was mandated at designated time points using enhanced computed tomography (CT) or magnetic resonance imaging (MRI). The images were copied onto CD-ROMs and shipped to the study coordinator. The study coordinator uploaded them to a cloud-based teleradiology system (Esite Healthcare Co. Ltd., Tokyo, Japan). Each panelist was provided with login accounts to view the full images on their computer terminals or other personal devices accessible to the Internet (Figure 1).

The surgical review was initially designed to be conducted voluntarily by the surgical expert panel in selected cases enrolled in the intermediate-risk group, using images obtained after two cycles of PLADO. Cases to be reviewed were selected by the surgical treatment coordinator of the study. Adding to these routine “voluntary reviews”, the panel responded to random surgical consultations from the treating institutions seeking recommendations for appropriate surgical treatment. Each review session was initiated with a review request e-mail from the study coordinator to all panelists. Personal and institutional information was anonymized, and case summaries, including AFP values, were sent along with detailed login information. To strictly avoid external access to the cloud system, and to protect the images from unintended use, IDs and passwords were managed centrally and changed periodically. Additionally, permission for access to the system was renewed annually according to the updated list of panelists. Panelists were allowed to browse the images online, but could not download the image files. The cloud-based viewer allowed all images, from diagnosis to the latest series, to be browsed through a single sequence. The panelists could easily compare side-by-side CT scans or MRI scans obtained at different time points, measure tumors, and change the contrast or brightness as they would do for patients in their facilities. The experts reviewed the images and made surgical decisions per protocol/surgical guideline. Opinions were collected via e-mail, and for cases in which consensus could not be reached through e-mail-based discussions, web meetings were held to reach an agreement.

The review results were summarized and integrated into a single report session by session. A final surgical review report was issued and sent to the treating institution. This report primarily provided recommendations on whether the institution should contact transplant centers early to prepare for transplantation. Further advice on surgical approaches or the content of discussion shared within the panelists were added as postscripts.

## 3. Results

### 3.1. Surgical Reviews and Responses to Consultations

A total of 41 surgical review sessions were conducted in 35 cases. Seven cases were subjected to two sessions at different time points. Six of the 35 cases were enrolled in the JPLT3-S study, 22 in the JPLT3-I, and 6 in the JPLT3-H. Of the 41 sessions, 27 were voluntary reviews, and 14 were held in response to consultations from treating physicians or surgeons (Table 1). Twenty-two out of the 28 voluntary reviews were performed on JPLT3-I (intermediate-risk) cases, since surgical review after two courses of PLADO was part of the study in this group, particularly for those with PRETEXT III and IV tumors.

### 3.2. Timing of Review Sessions

Regarding the timing of reviews, 74% (26 of 36) of the first review sessions were conducted using imaging studies obtained after one or two courses (blocks) of chemotherapy, which indicates that most of the reviews took place in the early phase of treatment (Table 2). Review sessions at later time points were mainly held in response to consultations from treating institutions, or as second reviews in cases that had undergone initial review sessions.

### 3.3. Promptness of Review Process

The median time consumed for surgical reviews was calculated as the number of days from the date of the actual imaging study to the date of issuing the surgical report. The average total session duration of all reviews was 21.9 days (Figure 2). When subclassified into voluntary reviews and consultation responses, the entire session duration was 23.4 days on average for voluntary reviews, and the average duration of consultation response was 19 days. The shorter time consumed in consultations may have reflected the nature of consultations, as these often require urgent answers.

To further clarify which part of the review process tends to consume more time, we next assessed the number of days from the imaging study to the beginning of the review session (period A), and from the beginning of the session to the issuing of the report (period B) (Figure 2). The exact date of starting sessions was not prospectively recorded; therefore, the date that the review request e-mail was sent was used as a surrogate. Nine of the 41 review sessions were excluded from this analysis because the corresponding e-mail could not be recovered. The average duration of period A was 14.8 days, compared to 9.3 days in period B. It is noteworthy that there was a wide range of days consumed in period A (2–64 days), whereas the actual reviewing process represented by period B was reasonably timely in all cases (1–20 days). The time spent for the actual review tended to increase when web conferences were used to reach consensus.

We next assessed whether the reports had been returned in time for the surgeons to plan their surgeries based on the recommendations. Thirty-two cases for which data on the date of surgery were available were subjected to this analysis. Regarding the first review session, the median duration from the date on which the surgical recommendation report was issued to the actual surgery date was 40 days (range: −1–98 days). The results were returned to the institutions more than 1 month before the surgery in 22 cases, whereas in the remaining 10 cases, the results were returned within a month prior to the surgery. In one urgent consultation case, the expert panel put out a recommendation as quickly as 7 days after the start of the review process, but the surgery had already been completed the day before the results were returned. The results of the first review session were reported on time for the operation in all other cases. In contrast to the first sessions, the second review sessions took place after four courses of chemotherapy in six cases, and after three courses in one (Table 2). The reports could be returned before the surgery in only two of the seven cases receiving second reviews, which indicates that reviews after four courses would likely not be in time to aid surgical decision-making.

### 3.4. Recommendations and Surgical Outcome/Patient Referral

As described previously, the primary aim of the surgical review was to predict whether the case may become a candidate for LTx, and to make recommendations for consulting a facility with expertise in complicated childhood liver surgery and transplantation. As a result of the first review session, 19 cases were predicted to become resectable with a conventional partial hepatectomy, including hemihepatectomy or trisectionectomy (Figure 3). For the other 16 cases, early consultation with a transplant center was recommended. Among these, the panel agreed that six cases were likely to require transplantation, even after further courses of chemotherapy. Ten cases were reported as “borderline” cases, in which the necessity of LTx was not definite but could not be excluded at the point of the review. Among these, there were cases in which the experts’ opinions varied and consensus for the recommended surgical approach could not be reached, including a case with a large POST-TEXT III tumor with multiple residual pulmonary nodules after induction chemotherapy; a case with tumor thrombus in the bilateral portal branches that were possibly resectable with portal reconstruction; or an infant case with a POST-TETX III tumor in which concerns were raised for insufficient residual liver volume. The decision for optimal surgical procedure was difficult for these cases even after extensive discussions through a web conference; thus, the final report concluded that the case requires early referral to transplant centers. The content of the discussion held within the expert panel was also reported to the treating institution for reference.

Of the 19 cases recommended to be treated with conventional partial hepatectomy, in addition to the two cases in which surgical data were missing, all 17 patients underwent hepatectomy either at the original institution or at a referring institution (Figure 3). Of the ten cases categorized into “borderline,” eight ultimately received hepatectomy either at the original institution or a transplant center, whereas two underwent transplantation. Of the six patients who were likely to require transplantation at the surgical review, four were ultimately treated with LTx, whereas two underwent hepatectomy.

Recommendations for consultation with transplant centers mandated neither patient referral nor the use of LTx, but aimed to facilitate the referral process once transplantation is needed. Nevertheless, 14 out of the 35 cases studied were transferred to a high-volume transplant center for liver surgery. These included two cases in which the surgical expert panel did not recommend consultation with a transplant center. For these two cases, the patients were referred at the discretion of the treating institutions. For the other 12 cases, we could not determine whether the surgical review report directly affected the institution to refer the patient, because data were not collected to clarify this. In contrast, there were three cases in the “borderline” group and one case in the “likely to require transplantation” group whose hepatectomies were ultimately performed in the original local institution (non-transplant centers). The decision on patient referral and the final surgical approach was left to the treating facility, and data on how these local centers decided to perform the surgery at their institution or whether the local centers had contacted transplant centers are unavailable.

## 4. Discussion

Standardization and quality assurance of protocol treatment are critical to the success of a multicenter clinical trial. From this perspective, compared to that of chemotherapy and radiotherapy, there are various obstacles to standardizing and assuring the quality of surgical treatment, including institutional variations in diagnostic imaging skills, variations in surgeons’ familiarity with the protocol, lack of uniformity in surgical approaches, unclear definition of acceptable surgeries, and lack of educational opportunities [18]. To overcome such obstacles, besides implicating a detailed surgical guideline, a real-time central surgical reviewing system was employed to support, or in some situations intervene with, the institutional decision-making process on time.

We deployed a cloud-based teleradiology system for central radiological reviewing in the current study. The technology for sharing medical images via the web has developed significantly in the last decade and is currently used for various medical purposes, including real-time remote radiological diagnosis [19,20]. Cloud-based remote image viewing systems serve as a diagnostic platform, and are ideal for central radiological reviews in a multicenter trial setting that requires accuracy and speed.

Regarding the timeliness of the process, since almost all review reports from the first review sessions were handed to the treating institutions before the actual surgical procedure, the rapidity of the reviews may be acceptable. However, patients with HB requiring referral to a transplant center would benefit from an even quicker response. The importance of early referral for transplant has recently been repeatedly emphasized by major collaborative study groups and transplant centers [9,21,22,23], and early consultation has been integrated as a secondary endpoint in the AHEP0731 study conducted by the Children’s Oncology Group [8,9,21]. An early referral does not necessarily mean that a transplant is imminent, but instead allows sufficient time for completing the transplant workup [8]. In our country, the vast majority of LTxs are living-donor transplantations [24], and even with this background, at least 1 month is required before surgery for donor selection and screening. Knowing that tumors can become drug-resistant after four or more courses of chemotherapy [25,26], it is widely recommended to avoid multiple courses before surgery. Thus, to facilitate the referral process and enable surgery, including LTx, after four to six courses of preoperative chemotherapy per protocol, the results of surgical reviews should be returned by the first day of the fourth course of chemotherapy. Our results showed that although many of the review sessions could be started within 2 weeks from the date of the imaging study, there were some cases in which period A (period from imaging study to initializing review) had been extraordinarily long. This was mostly caused by the delay of the treating institutions in sending the image data to the study coordinator. With further improvements in technology and systems, and educating treatment facilities on the significance of timeliness, we consider that this process could be shortened in future studies.

It is important to note that referral to a transplant center should not mandate LTx. There has been a continuous discussion on whether “borderline” unresectable tumors should ultimately be treated with transplantation or with extreme hepatectomies, without reaching a consensus to date [7,8,9,10,22,27,28,29]. Moreover, the resectability of the tumor could drastically improve in response to further courses of chemotherapy, enabling complete resection with hepatectomies free of margin for a tumor that once appeared to be unresectable. In addition, even in cases in which transplantation is unavoidable for the cure, donor factors or other social/familial issues must be considered for the indication of transplantation. For these reasons, in our real-time review system, the initial recommendation was limited to “consultation with” rather than “refer to” a transplant facility, expecting that with this recommendation, the local institution will contact a liver transplant facility and start mutual discussions on the indication for LTx.

This study had several limitations. First, because of its retrospective nature, data to determine whether surgical reviews directly influenced the surgical decisions at the treating institutions are unavailable. Second, this study focused on the practical operation of the web-based surgical review, and the detailed surgical procedures and outcomes, including long-term survival, recurrence, complications, and surgical margins, were not analyzed. These will be analyzed and published elsewhere once complete follow-up data become available.

## 5. Conclusions

In conclusion, a real-time surgical review system was adopted for the first time in a prospective multicenter clinical trial setting for HB. Our preliminary experience taught that critical recommendations on whether a patient should be consulted with a transplant center must be made using images obtained after two courses of chemotherapy at the latest. Although the direct effect of the reviews on surgical decision-making could not be evaluated, most of the reviewed cases promptly underwent definitive liver surgeries, including LTx, per protocol. The results suggest that central surgical reviews may facilitate centralization and benefit the quality assurance of surgical treatment in a multicenter setting. Real-time voluntary reviews on all patients undergoing preoperative chemotherapy in the current PHITT/JPLT-4 trial are currently conducted to verify our findings.

## Figures and Tables

**Figure 1 children-09-00234-f001:**
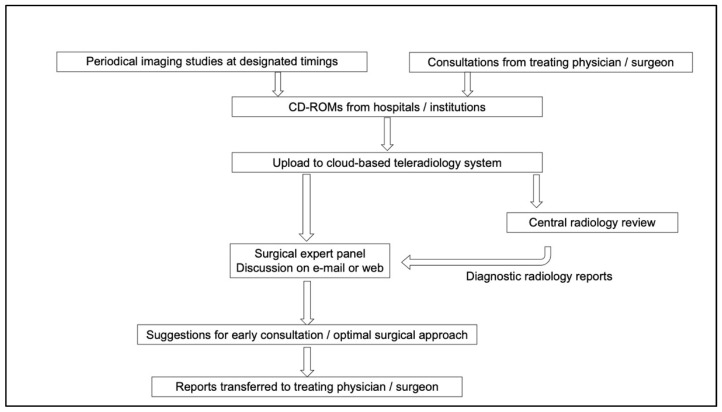
Flow of the central radiological and surgical reviews to facilitate early referral of enrolled patients in the The Japanese Study Group for Pediatric Liver Tumors 3 (JPLT3) study.

**Figure 2 children-09-00234-f002:**
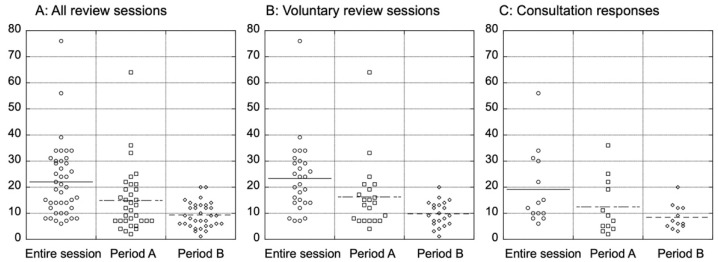
Duration of the first surgical review sessions in the studied 35 cases. “Entire session” indicates the number of days from the imaging study to the completion of the report. Period A indicates the number of days from the imaging study to the beginning of the reviewing process, and period B indicates the period from the beginning of the reviewing process to the issue of review report. (**A**) Duration of all review sessions included; (**B**) Duration of voluntary review sessions; (**C**) Duration of sessions in response to consultations from participating centers.

**Figure 3 children-09-00234-f003:**
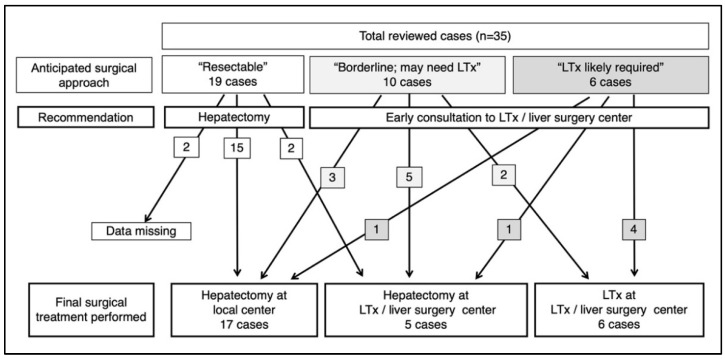
Recommendation of the expert panel at the first review session and surgical outcome of cases reviewed (JPLT3). Ltx: liver transplantation.

**Table 1 children-09-00234-t001:** Overview of review sessions.

	Total Enrollment	1st Session *n* = 35		2nd Session *n* = 7	
		voluntary	consultation	voluntary	consultation
standard	43	0	6	0	0
intermediate	36	20	2	6	1
high	15	2	5	0	0

**Table 2 children-09-00234-t002:** Timing of review sessions.

		1st Session *n* = 35	2nd Session *n* = 7
	after 2 courses	2	0
standard	after 4 courses	3	0
	after 5 courses	1	0
	after 2 courses	20	0
intermediate	after 3 courses	2	1
	after 4 courses	0	6
	after block A1	2	0
high	after block A2	2	0
	after block A3	3	0

## Data Availability

Research data are not shared.

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
