# Peer review of "Feasibility of Real-Time Central Surgical Review for Patients with Advanced-Stage Hepatoblastoma in the JPLT3 Trial"

_children, 2022, doi:10.3390/children9020234_

Round 1
Reviewer 1 Report
I want to congratulate the authors on this interesting approach in using a "real-time" surgical reviewing system for patients with hepatobastoma. I have no major concerns regarding this manuscript.
There are only minor questions that still remain for myself:
1) Why was there such a hetereogenicity in timing of reviews and as they were quicker in period B, what measures were implented to speed up the review process?
2) I am missing a statement concerning data privacy as you are using a web-based DICOM viewer system. Could you clarify how patient data was protected?
3) In paragraph 3.1. the author write "six cases were subjected to sessions twice at different timings", however Table 1 and Table 2 refer to 7 cases being reviewed twice. This would need clarification.
Author Response
We wish to express our appreciation to the Reviewer for his or her insightful comments, which have helped us significantly improve the paper. Please find our point-by-point response to your comments.
Point 1: Why was there such a hetereogenicity in timing of reviews and as they were quicker in period B, what measures were implented to speed up the review process?
Response 1: Thank you for your comment. The heterogeneity in period A accounted for such variation, which fundamentally reflected the inconsistency in the speediness of the institutions to download the images on CD-ROMs and sending them to the study center. We rewrote several sentences in the discussion to clearly explain this as follows.
“Our results showed that although many of the review sessions could be started within 2 weeks from the date of the imaging study, there were some cases in which period A (period from imaging study to initializing review) had been extraordinarily long. This was mostly caused by the delay of the treating institutions in sending the image data to the study coordinator.”
Point 2: I am missing a statement concerning data privacy as you are using a web-based DICOM viewer system. Could you clarify how patient data was protected?
Response 2: Thank you for your raising this important issue. We added a description referring to the technical solutions we applied to overcome leakiness of the system as follows,
“To strictly avoid external access to the cloud system and to protect the images from unintended use, IDs and passwords were managed centrally and changed periodically. Additionally, permission for access to the system was renewed annually according to the updated list of panelists. Panelists were allowed to browse the images online, but could not download the image files.”
Point 3: In paragraph 3.1. the author write "six cases were subjected to sessions twice at different timings", however Table 1 and Table 2 refer to 7 cases being reviewed twice. This would need clarification.
Response 3: The error is corrected in response to your query. The correct number is seven.
Reviewer 2 Report
The authors present an interesting article on their practice and methods of evaluation pediatric patients with high-risk hepatoblastoma in a multicenter setting. The paper is well written und it offers and important insight into the management of these difficult cases. The methods are described in adequate detail and the results are well presented. In addition to the already given information, it might have been interesting to present details on dissent or difficult discussions/decisions and how the decision process or recommendations were made in these cases.
All in all, this article offers important insights into the practice of evaluating high-risk hepatoblastoma patient within the JPLT-3 trial. Minor improvements of the language might be recommended.
Author Response
We wish to express our strong appreciation to the Reviewer for his or her insightful comments on our paper. We herein reply to the comments in a point-by -point manner.
Point 1: In addition to the already given information, it might have been interesting to present details on dissent or difficult discussions/decisions and how the decision process or recommendations were made in these cases.
Response 1: Thank you for your suggestion. We added several sentences to describe details of difficult cases and how we handled these, as shown below. Even with the established surgical guidelines, there are still many cases for which consensus is hard to be reached, and we feel that leading the institutions to refer the patient to transplant centers are sometimes the most that we can do.
“Among these, there were cases in which the experts’ opinions varied and consensus for the recommended surgical approach could not be reached, including a case with a large POST-TEXT III tumor with multiple residual pulmonary nodules after induction chemotherapy, a case with tumor thrombus in the bilateral portal branches that were possibly resectable with portal reconstruction, or an infant case with a POST-TETX III tumor in which concerns were raised for insufficient residual liver volume. The decision for optimal surgical procedure was difficult for these cases even after extensive discussions through a web conference; thus, the final report concluded that the case requires early referral to transplant centers. The content of the discussion held within the expert panel was also reported to the treating institution for reference.”
Point 2: Minor improvements of the language might be recommended.
Response 2: We had our manuscript brush up by a professional English editing service.